# Gene expression profiling of the masticatory muscle tendons and Achilles tendons under tensile strain in the Japanese macaque *Macaca fuscata*

Ko Ito[1], Yasuhiro Go[2,3,4], Shoji Tatsumoto[2], Chika Usui[2], Yosuke Mizuno[5], Eiji Ikami[6], Yuta Isozaki[1], Michihiko Usui[7], Takeshi Kajihara[8], Tetsuya Yoda[9], Ken-ichi Inoue[10], Masahiko Takada[10], Tsuyoshi Sato[1]*

1 Department of Oral and Maxillofacial Surgery, Saitama Medical University, Saitama, Japan, 2 Exploratory Research Center on Life and Living Systems (ExCELLS), National Institutes of Natural Science, Okazaki, Aichi, Japan, 3 Department of System Neuroscience, National Institute for Physiological Science, Okazaki, Aichi, Japan, 4 Department of Physiological Science, School of Life Science, SOKENDAI (The Graduate University for Advanced Studies), Okazaki, Aichi, Japan, 5 Division of Morphological Science, Biomedical Research Center, Saitama Medical University, Saitama, Japan, 6 Department of Oral and Maxillofacial Surgery, Hirosaki University Graduate School of Medicine, Hirosaki, Japan, 7 Division of Periodontology, Department of Cardiology and Periodontology, Kyushu Dental University, Fukuoka, Japan, 8 Department of Obstetrics and Gynecology, Saitama Medical University, Saitama, Japan, 9 Department of Maxillofacial Surgery, Graduate School of Medical and Dental Sciences, Tokyo Medical and Dental University, Tokyo, Japan, 10 Systems Neuroscience Section, Department of Neuroscience, Primate Research Institute, Kyoto University, Inuyama, Aichi, Japan

* tsato@saitama-med.ac.jp

**Data Availability Statement:** All relevant data are within the manuscript and its Supporting Information files.

## Abstract

Both Achilles and masticatory muscle tendons are large load-bearing structures, and excessive mechanical loading leads to hypertrophic changes in these tendons. In the maxillofacial region, hyperplasia of the masticatory muscle tendons and aponeurosis affect muscle extensibility resulting in limited mouth opening. Although gene expression profiles of Achilles and patellar tendons under mechanical strain are well investigated in rodents, the gene expression profile of the masticatory muscle tendons remains unexplored. Herein, we examined the gene expression pattern of masticatory muscle tendons and compared it with that of Achilles tendons under tensile strain conditions in the Japanese macaque *Macaca fuscata*. Primary tenocytes isolated from the masticatory muscle tendons (temporal tendon and masseter aponeurosis) and Achilles tendons were mechanically loaded using the tensile force and gene expression was analyzed using the next-generation sequencing. In tendons exposed to tensile strain, we identified 1076 differentially expressed genes with a false discovery rate (FDR) < $10^{-10}$. To identify genes that are differentially expressed in temporal tendon and masseter aponeurosis, an FDR of < $10^{-10}$ was used, whereas the FDR for Achilles tendons was set at > 0.05. Results showed that 147 genes are differentially expressed between temporal tendons and masseter aponeurosis, out of which, 125 human orthologs were identified using the Ensemble database. Eight of these orthologs were related to tendons and among them the expression of the glycoprotein nmb and sphingosine kinase 1 was increased in temporal tendons and masseter aponeurosis following exposure to tensile

**Funding:** Cooperative Research Program by PRI. JSPS KAKENHI Grant Number JP21H03137. The funder "Cooperative Research Program by PRI" had a role in study design, data collection and analysis. The funder "JSPS KAKENHI Grant Number JP21H03137" had a role in the decision to publish.

**Competing interests:** The authors have declared that no competing interests exist.

strain. Moreover, the expression of tubulin beta 3 class III, which promotes cell cycle progression, and septin 9, which promotes cytoskeletal rearrangements, were decreased in stretched Achilles tendon cells and their expression was increased in stretched masseter aponeurosis and temporal tendon cells. In conclusion, cyclic strain differentially affects gene expression in Achilles tendons and tendons of the masticatory muscles.

## Introduction

Tendons are elastic structures that transmit the force from muscles to the bone. The elasticity of tendons influences the function of the overall muscle-tendon complex. Tendons are composed of cells, including tenocytes, and extracellular matrix (ECM). The most abundant ECM protein in the tendon is type I collagen, while other ECM components include proteoglycans and glycoproteins. Small proteoglycans regulate collagen assembly while large proteoglycans resist compressive forces. Furthermore, glycoproteins help in maintaining lubrication and elasticity of tendons. The largest tendon in the human body is the Achilles tendon, which inserts into the calcaneus and is defined as the distal confluence of the gastrocnemius and soleus muscles anatomically [1]. In general, tendons are capable of adapting to mechanical loading. Rupture of the Achilles tendon is frequently observed during physical activity due to mechanical overloading. Thus, proper mechanical loading is essential for structural integrity and functions of the tendon, and affects the metabolism of the tendon tissue. Mechanical forces (i.e. tension, fluid shear stress, and compression) exert various biological effects on the tendon. It has been shown that cyclic tensile strain affects collagen synthesis [2]. Whereas short-term cyclic tensile strain inhibits collagen production, long-term cyclic tensile strain promotes collagen synthesis [3]. *In vitro*, tenocytes from tendon explants exhibit increased DNA synthesis and enhanced collagen production in response to cyclic tensile strain, and increased sulfated glycosaminoglycan (GAG) content in response to static loading [4].

Masticatory muscles in the maxillofacial region connect to maxillofacial bones via tendons and aponeuroses. Histologically, tendon entheses of the masticatory muscles resemble that of the limb skeleton [5]. Both the Achilles tendon and the masticatory muscle tendon are load bearing structures, and excessive mechanical loading on these tendons may lead to the development of hypertrophic changes [6, 7].

Masticatory muscle tendon-aponeurosis hyperplasia (MMTAH) is a condition in which the tendon and aponeurosis of the bilateral masticatory muscles exhibit hyperplasia that advances slowly after adolescence [8]. Hyperplasia of tendons and aponeuroses affects muscle extensibility, which results in a limited mouth opening. This condition is characterized by the presence of a palpable dense band at the anterior border of the masseter muscle visible upon maximum mouth opening and a square mandible configuration. Although the etiology of MMTAH remains unclear, clinical observations in individuals with parafunctional habits and juvenile bilateral onset suggest that environmental and genetic factors may be involved in the progression of this disease.

Histologically, tendons and aponeuroses in MMTAH appear normal due to the absence of inflammation and transformation. In our previous study, microstructural observation by electron microscopy revealed the presence of mineralized nodules of silicon along with calcium and phosphorus in tendons of patients with MMTAH but not in those with facial deformity [9]. Furthermore, using proteomic analysis, we observed that some proteins exhibit differential expression in tendons of patients with MMTAH but not in patients with facial deformity [10, 11]. Gene expression pattern of tenocytes isolated from the flexor, Achilles, and patellar tendons cultured under mechanical stress conditions is well investigated in rodents [12–14].

However, the gene expression profile of the masticatory muscle tendons remained unexplored. Herein, we investigated the gene expression profiles of the masticatory muscle tendons and Achilles tendons in the Japanese macaque *Macaca fuscata*. Clinical observations suggest that mechanical stress against masticatory tendons might be involved in MMTAH progression. In this study, mechanical loading was performed on primary tenocytes isolated from the masticatory muscle tendons and Achilles tendons of *M. fuscata*, and next-generation sequencing was performed to evaluate the gene expression profiles.

## Materials and methods

### Experimental animals and samples

This study was approved by the Animal Welfare and Animal Care Committee of the Primate Research Institute, Kyoto University. All experiments were performed in accordance with the Guide for Care and Use of Laboratory Primates issued by the institute (permit number, 2014–022). Adult male Japanese macaque *M. fuscata* (body weight, 8.0 kg) was housed in a cage under a 12–12 h light–dark cycle. The animal was fed a commercial monkey chow supplemented with fruits. Water was available *ad libitum*. The monkey was monitored closely and animal welfare was assessed on a daily basis and, if necessary, several times a day. This includes veterinary examinations to make sure that the animal is not suffering.

Since this monkey was used for other research projects (neural network tracing), tissue necessary for our study was obtained at the time of sacrifice. For euthanasia, the monkey was sedated with ketamine (8 mg/kg, i.m.) and medetomidine (0.04 mg/kg, i.m.), and then anesthetized deeply with an overdose of sodium pentobarbital (50 mg/kg, i.v.). After transcardial perfusion with 0.1 M phosphate-buffered saline (PBS, pH 7.4), the masseter aponeurosis (MA), temporal tendon (TT), and Achilles tendon (AT) were dissected. Finally, the monkey was perfused with 10% formalin in 0.1 M phosphate buffer (pH 7.4), and the brain was removed for histological analyses.

### Cell isolation procedure

Tenocytes were isolated according to Wagenhäuser's procedure [15]. Briefly, tendon slices were incubated in a 0.2% collagenase I solution (FUJIFILM Wako Pure Chemical Corporation, Osaka, Japan) for approximately 18 h at 37°C in a humidified environment with 5% $CO_2$. After digestion, cells were filtered by passing through a 100 μm cell strainer (BD Biosciences, Erembodegem, Belgium). The cells in the suspension were washed 3 times with PBS and centrifuged at 300 g for 5 min. Before initiating the cell culture, tenocytes were counted using a hemocytometer and trypan blue.

### Mechanical loading experiments

Cells were seeded at a density of $30 \times 10^4$ cells/well in a BioFlex 6-well plate (Flexcell International Corporation, Hillsborough, NC, USA). After 12 h of seeding, the cells were exposed to cyclic sinusoidal equi-biaxial tensile strain (from 0% to 10% amplitude at 1.0 Hz) for 48 h using the Flexcell FX-3000™ Tension System (Flexcell International Corporation) according to the manufacturer's instructions. Each experiment was performed in triplicate using three individual samples.

### RNA isolation and quality check

We compared the gene expression profiles of cells that were not exposed to the tensile strain (non-tension samples; 0 h) with those exposed to the tensile strain (tension samples, 48 h) (Table 1).

**Table 1. Samples for RNA-seq analysis.**

| tensile tension times | temporal tendon | masseter aponeurosis | Achilles tendon |
|---|---|---|---|
| 0 h | 3 replicates | 3 replicates | 3 replicates |
| 48 h | 3 replicates | 3 replicates | 3 replicates |

The harvested cells were rinsed with ice-cold PBS and total RNA was extracted using QIAzol (QIAGEN, Hilden, Germany). RNA purity was checked using the Nanodrop 2000® spectrophotometer (Thermo Fisher Scientific, MA, USA) and RNA concentration was measured using the Qubit® RNA Assay Kit and Qubit® 2.0 Fluorometer (Life Technologies, CA, USA). RNA integrity was assessed using the RNA Nano 6000 Assay Kit and the Agilent Bioanalyzer 2100 system (Agilent Technologies, CA, USA).

## RNA-sequencing

Libraries were prepared using the NEBNext Ultra Directional RNA Library Prep Kit for directional libraries (New England BioLabs) and the KAPA HTP Library Preparation Kit (KAPA Biosystems) according to the manufacturer's instructions. Sequencing was performed using the Illumina HiSeq platform. Raw sequence reads were aligned to the reference rhesus macaque genome (Mmul_8.0.1 in Emsembl) using HISAT2 (v2.1.0) [16]. The number of reads were counted using HTSeq (v0.9.1) [17] and expression levels were represented as fragments per kilobase of exon per million fragments mapped (FPKM) determined using StringTie (v1.3.4d) [18]. Finally, differentially expressed genes (DEGs) were identified using DESeq2 (v1.8.2) [19]. DEGs can be identified using a false discovery rate (FDR) of $< 0.05$; however, we applied the rigorous criteria (FDR $< 10^{-10}$) to identify genes that are differentially expressed in tenocytes following the application of tensile strain. Following criteria was used to identify DEGs: CON-A) DEGs among TT, MA, and AT samples were identified using the FDR of $< 10^{-10}$ and CON-B) DEGs between TT and MA groups (designated as the TM group) were identified using the FDR of $< 10^{-10}$ while FDR for AT was set at $> 0.05$.

## Results

To explore the differences in the gene expression pattern of masticatory tendons and Achilles tendons, we performed RNA-seq analysis of tenocytes isolated from these tendons. Results showed significant differences in the gene expression profile of cells before and after the application of tensile strain (Fig 1). We analyzed DEGs in un-stretched tendon cells (TT, MA, and AT) on the stipulation of CON-A. Compared with AT(0h), 1459 genes (833 up-regulated, 626 down-regulated) were differentially regulated in MA(0h) (S1 Table), whereas 1700 genes (1001 up-regulated, 699 down-regulated) were differentially regulated in TT(0h) (S1 Table). Compared with MA(0h), we found that 212 genes (127 up-regulated and 85 down-regulated) were differentially regulated in TT(0h) (S1 Table). Since hyperplasia is observed only in the tendon of masticatory muscles but not in the Achilles tendon of patients with MMTAH, we specifically focused on the masticatory muscle tendons. We analyzed DEGs in MA and TT on the stipulation of CON-B. Results showed that 1473 genes (794 up-regulated, 679 down-regulated) were differentially regulated between TT and MA at baseline (0h) (S2 Table).

Fig 2 summarizes the experimental design and comparison between different samples. Following the application of tensile strain (48 h vs 0 h), we identified 2195 DEGs in the TT group, 1914 DEGs in the MA group, and 3697 DEGs in the AT group. We also identified 1076 DEGs on the stipulation of CON-B (S3 Table), and 147 DEGs (39 up-regulated, 108 down-regulated) in the TM group (Fig 3; S4 Table).

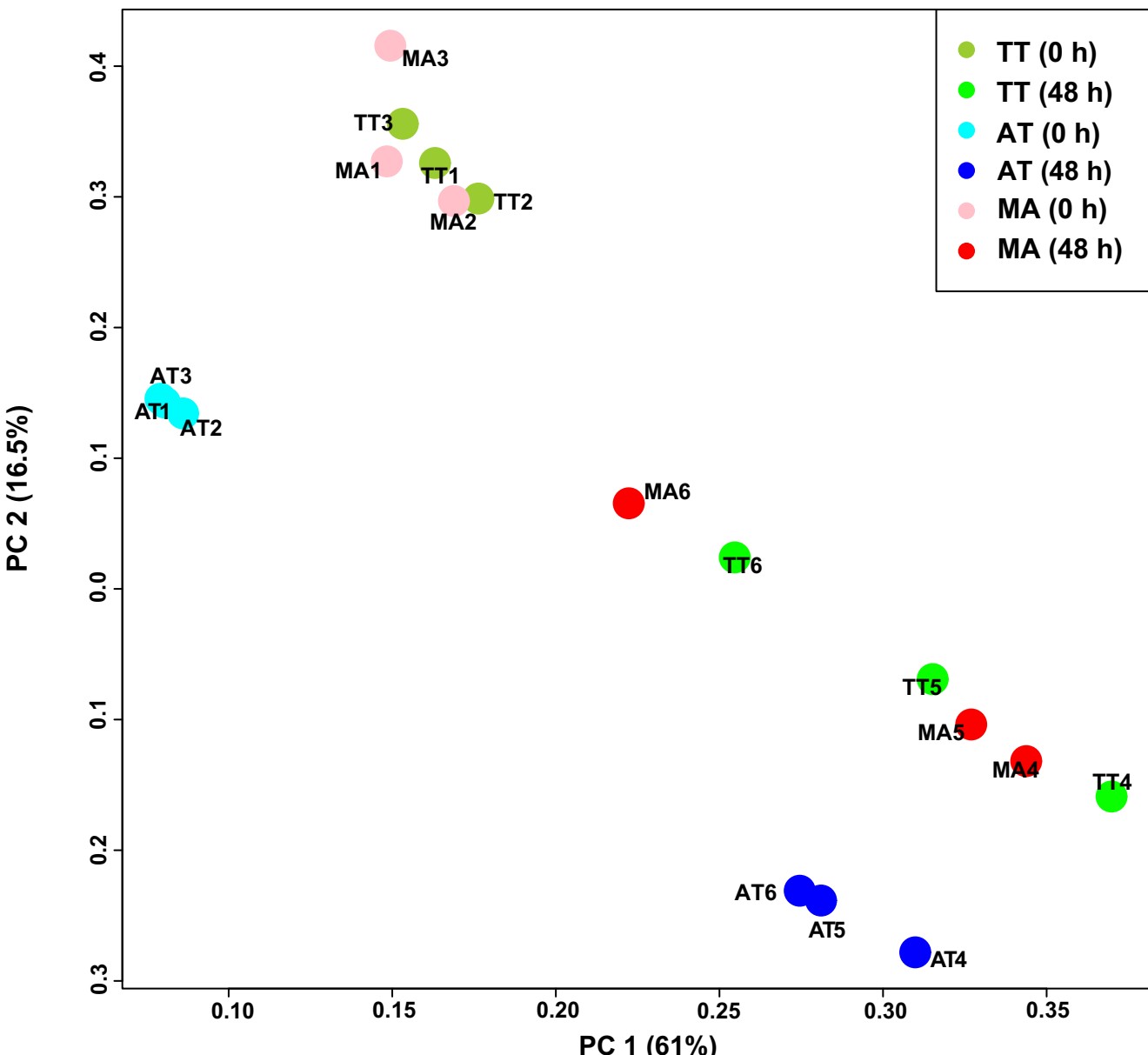

**Fig 1. Distinct gene expression pattern in each tendon sample.** Principal component analysis (PCA) of all samples.

Next, we performed functional analysis of the DEGs using the PANTHER GO-Slim online tool. Results showed that DEGs in the TM group were significantly enriched for the Gene Ontology (GO) terms cell cycle, division, fission, cytokinesis, and stress (cellular response to chemical stress and cellular response to oxidative stress) (Table 2).

GO annotation of 452 genes related to the cell cycle is shown in S4 Table. Fig 4 shows the hierarchical clustering of all samples. Among 147 DEGs that were differentially expressed only in TT and MA, we focused on 13 genes that were implicated in cell cycle regulation (S4 Table). Fig 5A shows the hierarchical clustering of these 13 genes, and their relative expression in all

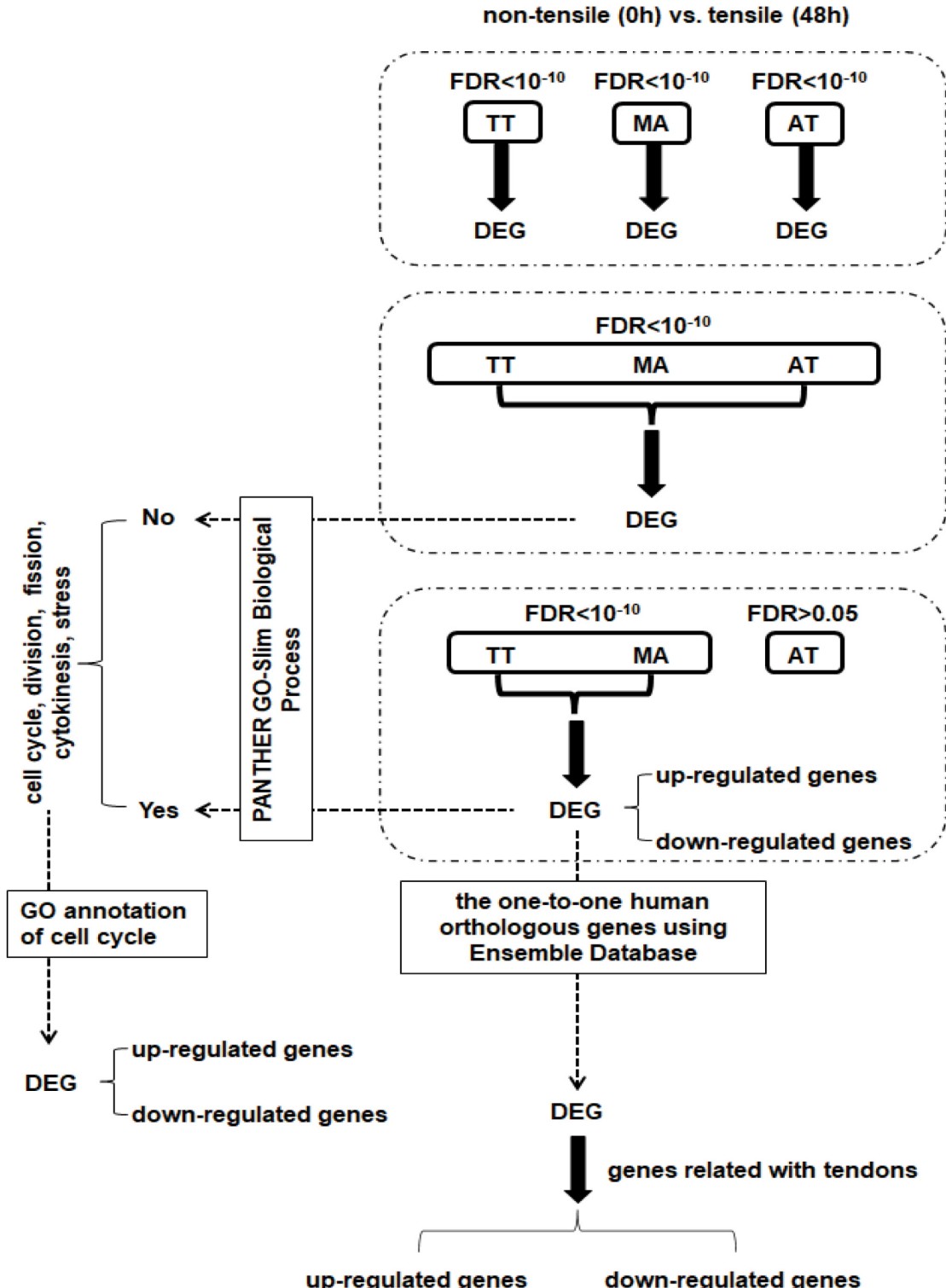

**Fig 2. Experimental design and scheme for comparing different samples.**

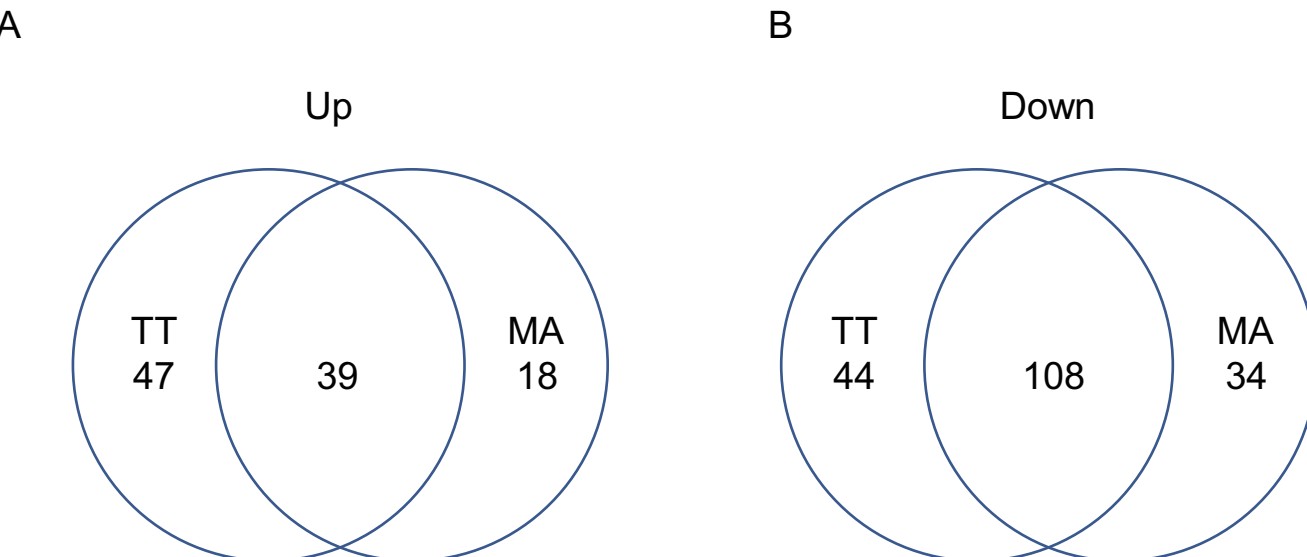

**Fig 3. DEGs in the TM group.** Venn diagram showing up- or down- regulated genes in the TM group. The number of up-regulated genes are less than that of the down-regulated genes.

samples is shown in Fig 5B. Among these 13 genes, polo like kinase 2 (*PLK2*) and tubulin beta 3 class III (*TUBB3/TUJ1*) are related to tendons [20–22]. The expression of septin 9 (*SEPTIN9*) and *TUBB3* were increased, whereas the expression of anillin, actin binding protein (*ANLN*),

**Table 2. Gene ontology (GO) analysis of 1076 DEGs in all tendons (TT, MA, and AT) and 147 DEGs in TT and MA.**

| PANTHER GO-Slim Biological Process | *Macaca mulatta* REFLIST (21794) | 1076 DEGs (AT/TT/ MA < 10⁻¹⁰) FDR | 147 DEGs (AT > 0.05 & TT/ MA < 10⁻¹⁰) FDR |
|---|---|---|---|
| cell cycle (GO:0007049) | 447 | ND | 1.34E-03 |
| cell cycle process (GO:0022402) | 405 | ND | 1.41E-03 |
| organelle fission (GO:0048285) | 318 | ND | 4.47E-03 |
| nuclear division (GO:0000280) | 296 | ND | 3.43E-03 |
| mitotic cell cycle (GO:0000278) | 251 | ND | 2.80E-03 |
| mitotic cell cycle process (GO:1903047) | 251 | ND | 2.24E-03 |
| mitotic nuclear division (GO:0140014) | 251 | ND | 1.87E-03 |
| cell division (GO:0051301) | 59 | ND | 5.10E-03 |
| membrane fission (GO:0090148) | 56 | ND | 4.25E-03 |
| cytokinesis (GO:0000910) | 47 | ND | 3.07E-03 |
| cytoskeleton-dependent cytokinesis (GO:0061640) | 32 | ND | 6.59E-03 |
| cellular response to chemical stress (GO:0062197) | 21 | ND | 2.62E-02 |
| cellular response to oxidative stress (GO:0034599) | 20 | ND | 2.44E-02 |

AT/TT/MA; AT, TT and MA

TT/MA; TT and MA

ND; not detected

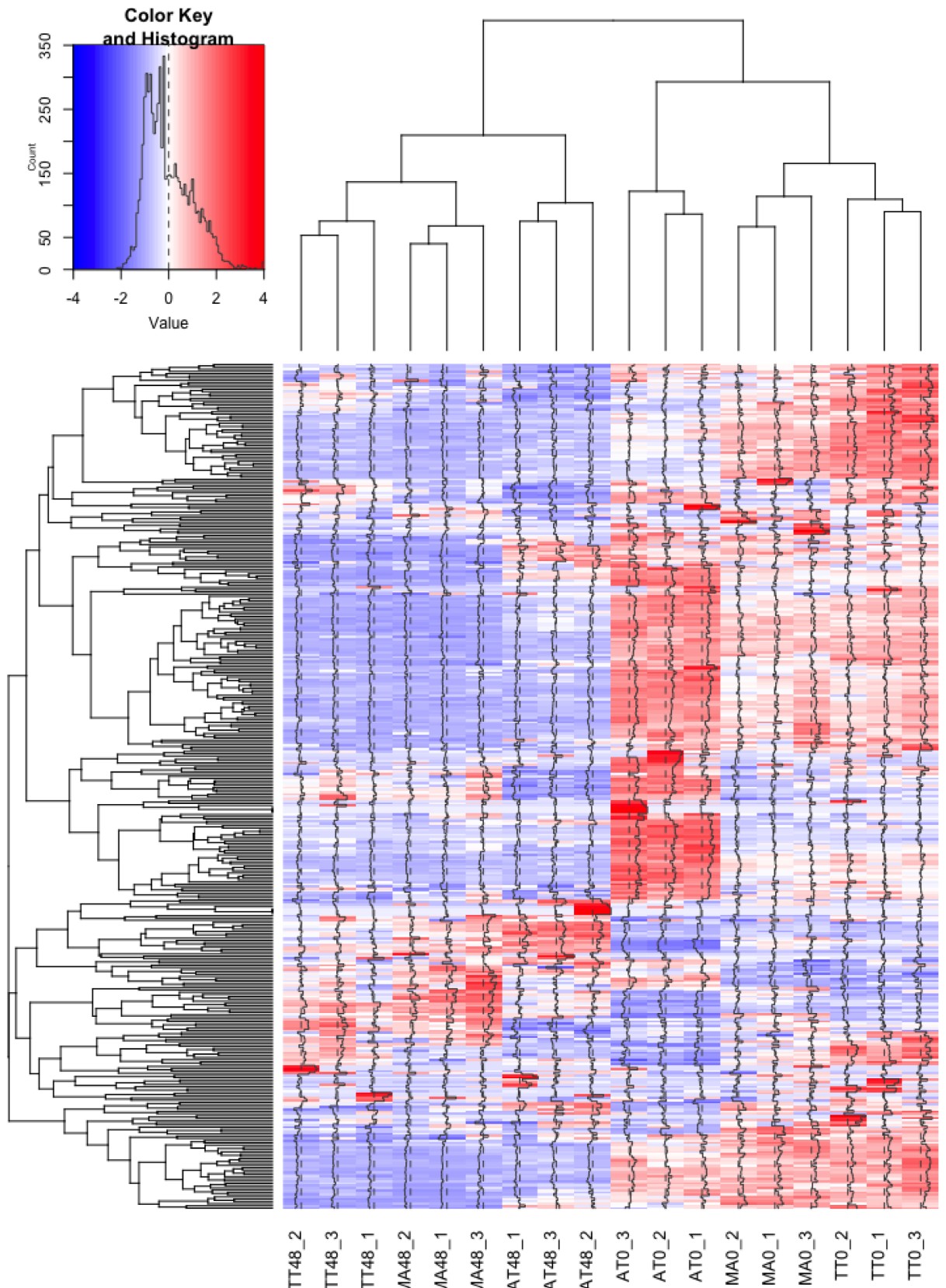

**Fig 4. Gene expression profile of each tendon sample.** Heatmap with hierarchical clustering of the 452 DEGs in TT, MA, and AT samples (red, up-regulated; blue, down-regulated). Histogram shows the numerical value of each color.

annexin A1 (*ANXA1*), mitotic checkpoint protein (*BUB3*), cyclin G1 (*CCNG1*), epithelial cell transforming 2 (*ECT2*), cyclin-dependent kinase 1 (*CDK1*), PDS5 cohesin associated factor B (*PDS5B*), polo like kinase 2 (*PLK2*), septin 11 (*SEPTIN11*), and DNA topoisomerase II alpha (*TOP2A*) was decreased in tendon cells exposed to tensile strain.

Next, we examined the one-to-one human orthologs of these 147 DEGs using the Ensemble database and identified 125 such orthologs (S5 Table). Among these orthologs, eight genes were related to tendons (Table 3), including sphingosine kinase 1 (*SPHK1*), alpha-2-macroglobulin (*A2M*), glycoprotein nmb (*GPNMB*), slit guidance ligand 3 (*SLIT3*), early growth response protein 1 (*EGR1*), ADAM metallopeptidase with thrombospondin type 1 motif 12 (*ADAMTS12*), *CDK1*, and follistatin-like 1 (*FSTL1*). The expression of *GPNMB* and *SPHK1* was increased, while the expression of the other six genes was decreased in tenocytes following the application of tensile strain (Fig 6).

## Discussion

Although RNA-seq analysis of Achilles and patellar tendons have been performed in mice [23], studies comparing the gene expression profiles of Achilles and masticatory muscle tendons are sparse. Herein, we observed that the number of genes that were differentially regulated between AT and TT or between AT and MA were much more than the number of DEGs between MA and TT (S1 Table), suggesting that the characteristics of the Achilles tendon is different from that of the masticatory muscle tendon. To the best of our knowledge, our study

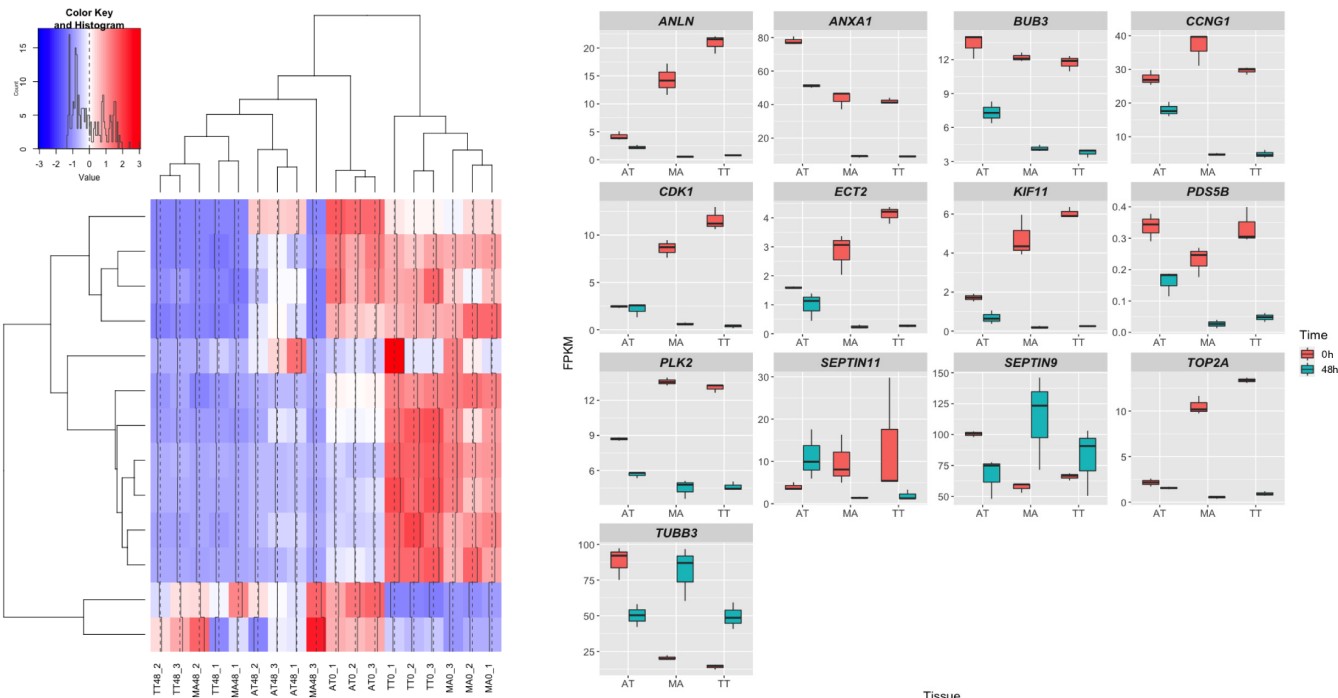

**Fig 5. Heatmap and boxplot showing the expression of 13 genes.** A) Heatmap with hierarchical clustering of 13 DEGs (red, up-regulated; blue, down-regulated). Histogram shows the numerical values of each color. B) Boxplot showing the relative expression of 13 selected DEGs.

**Table 3. Tendon-related genes among 147 DEGs in TT and MA.**

| Gene Symbol | Description |
| --- | --- |
| FSTL1 | follistatin like 1 |
| A2M | alpha-2-macroglobulin |
| GPNMB | glycoprotein nmb |
| ADAMTS12 | ADAM metallopeptidase with thrombospondin type 1 motif 12 |
| SLIT3 | slit guidance ligand 3 |
| SPHK1 | sphingosine kinase 1 |
| EGR1 | early growth response 1 |
| CDK1 | cyclin-dependent kinase 1 |

is the first to compare the gene expression profiles of the Achilles tendon and tendons of the masticatory muscles in the Japanese macaque using next-generation sequencing. Our previous study showed that cyclic tensile strain induces decorin expression via yes-associated protein in cultured tenocytes and tendons of patients with MMTAH, suggesting that in MMTAH,

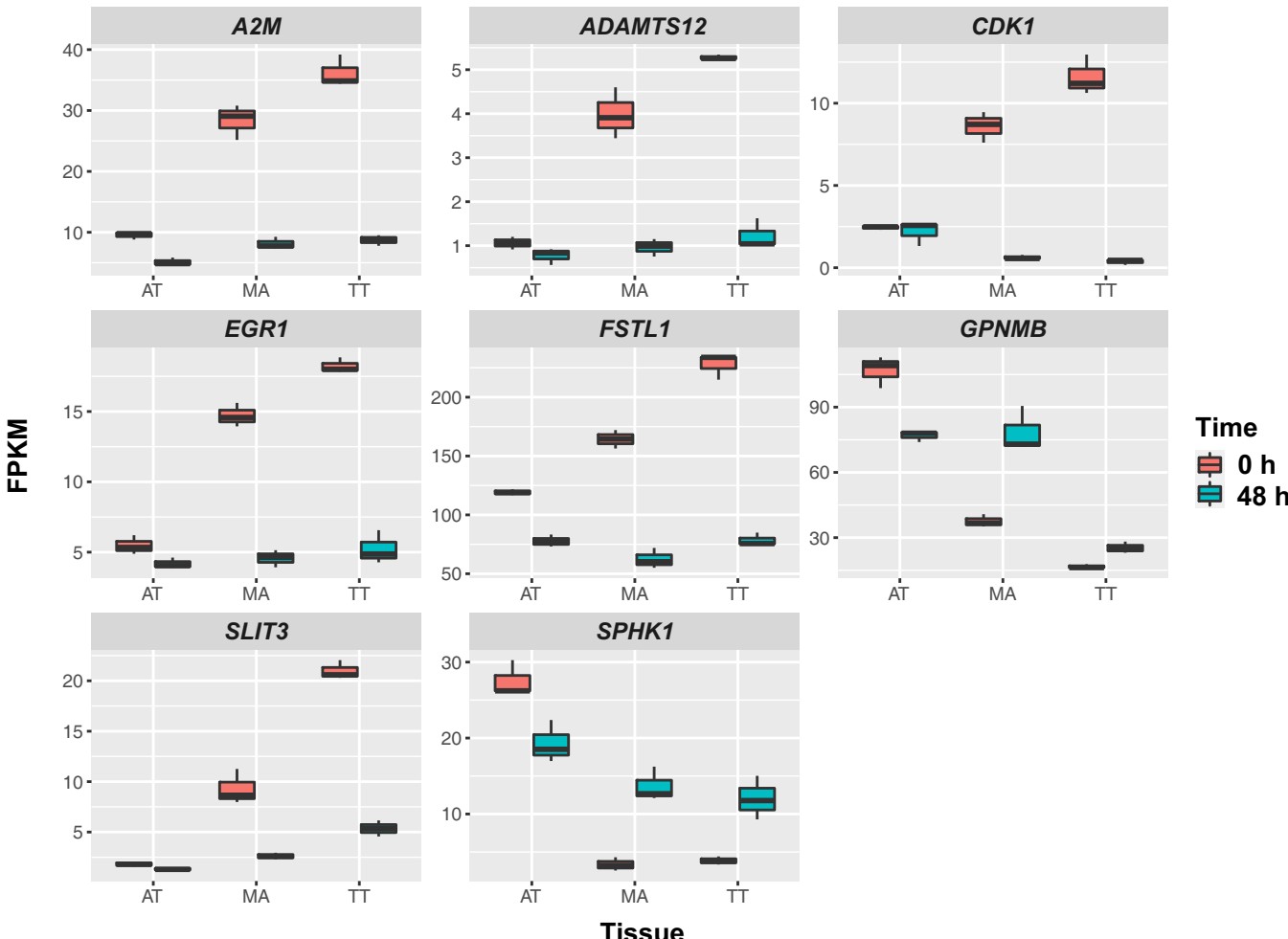

**Fig 6. Changes in the expression of the eight tendon-related genes after 48 h of exposure to tensile strain in AT, MA, and TT.** The expression of *GPNMB* and *SPHK1* is up-regulated whereas that of *A2M*, *ADAMTS12*, *CDK1*, *EGR1*, *FSTL1*, and *SLIT3* is down-regulated in TT and MA following tensile strain.

tendons are subjected to cyclic tensile strain [24]. Since DEGs in the tendon of masticatory muscles under tensile strain were enriched for the GO term division, it can be speculated that tensile strain promotes cell proliferation in these tendons.

The effects of cyclic tensile strain on tendon cells are well investigated. Although various strain amplitudes have been tested, most experiments used the frequency of 1.0 Hz, and very few studies have used the cyclic strain at low amplitude (1–3%). Lavagnino *et al.* reported that 1% static strain at a frequency of 1.0 Hz for 24 h inhibits matrix metalloproteinase (MMP)-1 expression in rat tail tendon fascicles *in vitro* [25]. Furthermore, a cyclic strain of 3% amplitude superimposed on a 2% static strain for 24 h enhances the retention of newly synthesized collagen into the fascicle of the rat tail tendon [3]. Another study by Lavagnino *et al.* [26] demonstrated that biaxial cyclic tensile strain of 12% amplitude at 1.0 Hz for 24 h significantly decreases actin depolymerization and increases collagenase expression. Based on these studies, we used cyclic sinusoidal equi-biaxial tensile strain at 1.0 Hz for 48 h from low (0%) to high (10%) amplitude in tendons isolated from the Japanese macaque.

Maeda *et al.* performed microarray analysis of tenocytes exposed to cyclic strain [27]. In rat tail tendon fascicles, cyclic strain of 3% amplitude superimposed on a 2% static condition significantly reduces the expression of some MMP and ADAMTS family proteins while increases the expression of types VI and VIII collagen. This suggests that a cyclic strain of 3% amplitude superimposed on a 2% static strain for 24 h has anabolic effects on tenocytes.

A previous study showed that ruptured Achilles tendons are surrounded by *TUBB3* positive cells [21]. *TUBB3* is expressed in non-neuronal tumor cells and plays a key role in regulating cell cycle progression [28]. Moreover, *SEPTIN9* promotes cell migration and cytoskeletal rearrangements [29]. Our results showed that both *TUBB3* and *SEPTIN9* mRNA levels were decreased in stretched AT cells while their levels were increased in stretched MA and TT cells (Fig 6). We speculate that cyclic tensile strain induces cell cycle progression and cytoskeletal rearrangements in masticatory muscle tendon cells.

In our study, cyclic tensile strain induced the expression of both *SPHK1* and *GPNMB* in tenocytes derived from the tendon of masticatory muscles (Fig 3). Intriguingly, Mousavizadeh *et al.* reported that the expression of *SPHK1* is up-regulated by cyclic strain in human tendon cells [30]. Since *GPNMB* up-regulation is associated with tendon regeneration [31], it is possible that tenocytes of masticatory muscles have a high regenerative capability.

The expression levels of both *SLIT3* and *EGR1*, which are essential for tenogenic differentiation [32, 33] were found to be decreased, indicating that tenocytes from the masticatory muscle tendons switch from the differentiation state to the proliferation state under cyclic strain. Our study revealed that the expression of *ADAMTS12*, which prevents heterotopic ossification in tendons is down-regulated in tenocytes under tensile strain [34], which is consistent with the results of a previous microarray-based study on rat tail tendons under cyclic strain [27]. In our previous study, we observed heterotopic calcification in tendons of the masticatory muscles in primates with MMTAH [9]; however, this study was performed in a single species. We speculate that cyclic strain may promote heterotopic calcification in tendons of the masticatory muscles accompanied with the down-regulation of *ADAMTS12* expression. In contrast, the expression of *A2M* and *CDK1*, which regulates regeneration and proliferation in rotator cuff tendons and Achilles tendons was decreased (Fig 3), suggesting that *A2M* and *CDK1* are not essential for proper functioning of the masticatory muscle tendons [35, 36]. Although *FSTL1* is highly expressed in tendons [37], we found that the expression of *FSTL1* is lower in the tendon of masticatory muscles. These results suggest that cyclic sinusoidal equi-biaxial tensile strain has anabolic effects on tenocytes.

Normally, the force applied to the masticatory tendon during chewing is approximately 320 N, and the force applied to the Achilles tendon during walking is approximately 1,000 N [38, 39].

It is assumed that the Achilles tendon has high load-bearing properties. On the other hand, it has been reported that patients with MMTAH show hyperplasia of the masticatory muscle tendon without hyperplasia of the Achilles tendon, but the masticatory muscle tendon is subjected to a bite force of approximately 1,000 N [40]. Given these facts, it is consistent to assume that different genes are expressed in masticatory muscle tendons, which are considered vulnerable to high loads, than in Achilles tendons, which can withstand high loads. Although high loads were applied to each tendon cell in this study, different genes were expressed in the masticatory muscle tendon and Achilles tendon, respectively. In masticatory muscle tendon cells, which are vulnerable to high loads, the expression of genes that promote cell cycle progression and cytoskeletal reorganization (*TUBB3* and *SEPTIN9*) and genes important for tendon regenerative capacity (*SPHK1* and *GPNMB*) were elevated. We speculate that the faster turnover in highly loaded masticatory muscle tendon cells may increase the speed of cell regeneration, resulting in hyperplasia of tendon tissue. In other words, differences in gene expression between masticatory muscle tendons and Achilles tendons may be associated with differences in vulnerability to loading.

Our study has some limitations. First, since only one monkey was used in our study, our results cannot be generalized. Second, the possibility that our tendon samples may contain the muscle tissue at the micro level cannot be excluded. Nevertheless, our study is meaningful because no previous study has performed next-generation sequencing to compare the transcriptomes of the Achilles tendons and tendons of the masticatory muscles.

## Conclusions

Cyclic strain differentially affects gene expression in Achilles tendons and tendons of the masticatory muscles. In the future, we have planned to investigate whether *SPHK1* and *GPNMB* are highly expressed in tendon and blood samples of individuals with MMTAH, and whether transgenic mice without functional *SPHK1* and *GPNMB* genes exhibit the clinical phenotype of MMTAH.

## Supporting information

**S1 Table. Differentially expressed genes in un-stretched TT, MA, and AT cells on the stipulation of CON-A.**
(XLSX)

**S2 Table. Differentially expressed genes of the TM group on the stipulation of CON-B.**
(XLSX)

**S3 Table. Differentially expressed genes with the condition of FDR $< 10^{-10}$ in TT, MA, and AT.**
(XLSX)

**S4 Table. Differentially expressed genes with the condition of FDR $< 10^{-10}$ for TT and MA and FDR $> 0.05$ for AT.**
(XLSX)

**S5 Table. List of human one-to-one orthologs differentially expressed only in TT and MA.**
Thirty-nine genes were up-regulated and one hundred eight genes were down-regulated.
(XLSX)

## Acknowledgments

I would like to thank Professor Dr. Tetsuo Ikezono (Department of Otorhinolaryngology, Saitama Medical University) because he introduced Masahiko Takada to Tsuyoshi Sato. We would like to thank Editage (www.editage.com) for English language editing.

## Author Contributions

**Conceptualization:** Tsuyoshi Sato.

**Data curation:** Yasuhiro Go, Yosuke Mizuno, Tsuyoshi Sato.

**Formal analysis:** Ko Ito, Yasuhiro Go, Yosuke Mizuno.

**Funding acquisition:** Masahiko Takada.

**Investigation:** Ko Ito, Shoji Tatsumoto, Chika Usui, Eiji Ikami, Yuta Isozaki, Takeshi Kajihara.

**Project administration:** Tsuyoshi Sato.

**Resources:** Ken-ichi Inoue, Masahiko Takada.

**Supervision:** Tetsuya Yoda.

**Visualization:** Yasuhiro Go, Yosuke Mizuno.

**Writing – original draft:** Ko Ito.

**Writing – review & editing:** Yasuhiro Go, Michihiko Usui, Tsuyoshi Sato.

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
