## [Decision Letter · Decision Letter 0]

30 May 2022

PONE-D-22-06866Gene expression analysis of masticatory tendon and Achilles tendon with tensile tension in a Japanese monkey by using RNA-sequencingPLOS ONE

Dear Dr. Sato,

Thank you for submitting your manuscript to PLOS ONE. After careful consideration, we feel that it has merit but does not fully meet PLOS ONE’s publication criteria as it currently stands. Therefore, we invite you to submit a revised version of the manuscript that addresses the points raised during the review process.

Please answer the 5 points raised by the reviewer.- clarify the result section as suggested- include include baseline comparisons between all three tendons- include a heatmap of the genes that are implicated in the cell cycle GO categories- move the FDR criteria and comparisons in the methods section- please use English Proofreading 

We look forward to receiving your revised manuscript.

Kind regards,

Claude Prigent

Academic Editor

PLOS ONE

Journal Requirements:

- https://www.ijoms.com/article/S0901-5027(13)00267-1/fulltext

- https://onlinelibrary.wiley.com/doi/10.1111/odi.12876

The text that needs to be addressed involves the Introduction

In your revision ensure you cite all your sources (including your own works), and quote or rephrase any duplicated text outside the methods section. Further consideration is dependent on these concerns being addressed.

“Cooperative Research Program by PRI.

JSPS KAKENHI Grant Number JP21H03137.”

“This work was supported by the Cooperative Research Program by PRI (for TS) and partly by JSPS KAKENHI Grant Number JP21H03137 (for TY)”

“Cooperative Research Program by PRI.

JSPS KAKENHI Grant Number JP21H03137.”

Reviewers' comments:

Reviewer's Responses to Questions

**Comments to the Author**

1. Is the manuscript technically sound, and do the data support the conclusions?

Reviewer #1: Partly

2. Has the statistical analysis been performed appropriately and rigorously? 

Reviewer #1: I Don't Know

3. Have the authors made all data underlying the findings in their manuscript fully available?

Reviewer #1: Yes

4. Is the manuscript presented in an intelligible fashion and written in standard English?

Reviewer #1: Yes

5. Review Comments to the Author

Reviewer #1: In this study, the authors have identified distinct tendon-specific mechanoresponsive transcriptional changes in primary tendon cells in the Japanese monkey (Macaca fuscata) from three different tendons: the Achilles tendon, temporal tendon and masseter aponeurosis. In particular, they have determined that maxillofacial tendons exhibit more differentially expressed genes pertaining to cell proliferation and stress compared to Achilles tendon in response to loading. These findings may help provide insight into the pathogenesis of masticatory muscle tendon–aponeurosis hyperplasia (MMTAH), a condition that is prevalent in masticatory tendons but not Achilles tendon. In addition, this work is one of the first to profile the difference in transcriptional responses between limb vs. craniofacial tendons in response to mechanical loading in primates and as such, would be of interest to the tendon field. However, substantial revisions need to be performed prior to publication.

Major comments:

1. The text in the results section should be revised for clarity purposes. With multiple tendons for each condition, it should be clearly stated what comparisons were being made to arrive at the differential gene expression list. We also suggest including a graphical figure clearly diagramming the experimental design and RNA-seq comparisons made between the tendons/conditions. Some (but not all) specific text examples we refer the authors to for revision are listed below:

-Page 7 Lines 18 - 20

We identified 2195 genes in temporal tendon (TT), 1914 genes in masseter aponeurosis (MA) and 3697 genes in Achilles tendon (AT) as differentially expressed genes (DEGs) with false discovery rate (FDR) < 10-10.

-Page 7 Lines 20 - Page 8 Line 1

We also identified 1076 DEGs that meet the following condition; FDR < 10-10 in all tendons (TT, MA, and AT) (S1 Table).

-Page 8 Lines 1 - 4

By contrast, as the hyperplasia is observed in tendon of masticatory muscles not but in Achilles tendon in MMTAH patients, we examined DEGs that meet the following condition; TT and MA (designated as TM group) for FDR<10-10 and AT for FDR>0.05.

2. There appear to be substantial differences between the AT and MA/TT tendons at baseline (no stretching) which is interesting. These differences at baseline may provide insight into the differential responses between AT and MA/TT upon stretching. As such, the authors should include this baseline comparison data analysis between all three tendons and add it to the manuscript.

3. It would be nice to include a heatmap of the genes that are implicated in the cell cycle GO categories and also to see their relative expression levels compared to Achilles tendon conditions.

4. The FDR criteria and exact comparisons made for differential expression should be included in the methods section. The FDR criteria in particular can be moved into the methods section completely for clarity in the results section of the text.

5. The writing quality needs to be improved significantly throughout the manuscript. The Introduction and Discussion sections are extremely short and superficially written. They should be revised to include more detail about previous works and implications of these works both in context not just of their own work, but other studies in the tendon field.

Minor comments:

Page 7, Line 16: The word “ingredient” is confusing here and should be removed. Perhaps rephrase the sentence to clearly indicate to the reader that tendon-specific trends in gene expression were analyzed with respect to stretching?

Page 12, line 16: “MMATH” should be changed to “MMTAH”

Page 12, line 16: The use of the word “primate” is preferred here. In addition, the authors should clarify whether they are referring to one species of primate or other monkeys of the same species.

6. PLOS authors have the option to publish the peer review history of their article (what does this mean?). If published, this will include your full peer review and any attached files.

Reviewer #1: No

---

## [Author Response · Author response to Decision Letter 0]

16 Oct 2022

Response to reviewers

We thank the Editor and the Reviewers for their insightful and helpful comments. We have tried to address all issues raised by the reviewers. Also, we have thoroughly revised the manuscript and adding new figures and tables. We highlighted the corrected or additional words/sentences with red font and the proofread words/sentences with the underline.

Please answer the 5 points raised by the reviewer.

- clarify the result section as suggested

- include baseline comparisons between all three tendons

- include a heatmap of the genes that are implicated in the cell cycle GO categories

- move the FDR criteria and comparisons in the methods section

- please use English Proofreading

Review Comments to the Author

Reviewer #1: In this study, the authors have identified distinct tendon-specific mechanoresponsive transcriptional changes in primary tendon cells in the Japanese monkey (Macaca fuscata) from three different tendons: the Achilles tendon, temporal tendon and masseter aponeurosis. In particular, they have determined that maxillofacial tendons exhibit more differentially expressed genes pertaining to cell proliferation and stress compared to Achilles tendon in response to loading. These findings may help provide insight into the pathogenesis of masticatory muscle tendon–aponeurosis hyperplasia (MMTAH), a condition that is prevalent in masticatory tendons but not Achilles tendon. In addition, this work is one of the first to profile the difference in transcriptional responses between limb vs. craniofacial tendons in response to mechanical loading in primates and as such, would be of interest to the tendon field. However, substantial revisions need to be performed prior to publication.

Major comments:

1. The text in the results section should be revised for clarity purposes. With multiple tendons for each condition, it should be clearly stated what comparisons were being made to arrive at the differential gene expression list. We also suggest including a graphical figure clearly diagramming the experimental design and RNA-seq comparisons made between the tendons/conditions. Some (but not all) specific text examples we refer the authors to for revision are listed below:

-Page 7 Lines 18 - 20

We identified 2195 genes in temporal tendon (TT), 1914 genes in masseter aponeurosis (MA) and 3697 genes in Achilles tendon (AT) as differentially expressed genes (DEGs) with false discovery rate (FDR) < 10-10.

-Page 7 Lines 20 - Page 8 Line 1

We also identified 1076 DEGs that meet the following condition; FDR < 10-10 in all tendons (TT, MA, and AT) (S1 Table).

-Page 8 Lines 1 - 4

By contrast, as the hyperplasia is observed in tendon of masticatory muscles not but in Achilles tendon in MMTAH patients, we examined DEGs that meet the following condition; TT and MA (designated as TM group) for FDR<10-10 and AT for FDR>0.05.

Author’s Response: Thank you very much for the important suggestions. We have added a statement to clarify the purpose of the study in the Results section (line 179–181): “To explore the differences in the gene expression pattern of masticatory tendons and Achilles tendons, we performed RNA-seq analysis of tenocytes isolated from these tendons.” Furthermore, we have also added a figure (Figure 2) showing the experimental design and comparison along with description of the figure in the Results section (line 184): “Fig 2 summarizes the experimental design and comparison between different samples.” 

 

2. There appear to be substantial differences between the AT and MA/TT tendons at baseline (no stretching) which is interesting. These differences at baseline may provide insight into the differential responses between AT and MA/TT upon stretching. As such, the authors should include this baseline comparison data analysis between all three tendons and add it to the manuscript.

Author’s Response: Thank you for your insightful suggestions. We have added new Tables (S1 and S2) and the following details (line 182–192): “We analyzed DEGs in un-stretched tendon cells (TT, MA, and AT) on the stipulation of CON-A. Compared with AT(0h), 1459 genes (833 up-regulated, 626 down-regulated) were differentially regulated in MA(0h) (S1 Table), whereas 1700 genes (1001 up-regulated, 699 down-regulated) were differentially regulated in TT(0h) (S1 Table). Compared with MA(0h), we found that 212 genes (127 up-regulated and 85 down-regulated) were differentially regulated in TT(0h) (S1 Table). Since hyperplasia is observed only in the tendon of masticatory muscles but not in the Achilles tendon of patients with MMTAH, we specifically focused on the masticatory muscle tendons. We analyzed DEGs in MA and TT on the stipulation of CON-B. Results showed that 1473 genes (794 up-regulated, 679 down-regulated) were differentially regulated between TT and MA at baseline (0h) (S2 Table).”

3. It would be nice to include a heatmap of the genes that are implicated in the cell cycle GO categories and also to see their relative expression levels compared to Achilles tendon conditions.

Author’s Response: We have added new figures (Figure 4, 5A, and 5B) and the following text in the Results section (line 216–234). “GO annotation of 452 genes related to the cell cycle is shown in S4 Table. Fig 4 shows the hierarchical clustering of all samples. Among 147 DEGs that were differentially expressed only in TT and MA, we focused on 13 genes that were implicated in cell cycle regulation (S4 Table). Fig 5A shows the hierarchical clustering of these 13 genes, and their relative expression in all samples is shown in Fig 5B. Among these 13 genes, polo like kinase 2 (PLK2) and tubulin beta 3 class III (TUBB3/TUJ1) are related to tendons [20-22]. The expression of septin 9 (SEPTIN9) and TUBB3 were increased, whereas the expression of anillin, actin binding protein (ANLN), annexin A1 (ANXA1), mitotic checkpoint protein (BUB3), cyclin G1 (CCNG1), epithelial cell transforming 2 (ECT2), cyclin-dependent kinase 1 (CDK1), PDS5 cohesin associated factor B (PDS5B), polo like kinase 2 (PLK2), septin 11 (SEPTIN11), and DNA topoisomerase II alpha (TOP2A) was decreased in tendon cells exposed to tensile strain.” 

4. The FDR criteria and exact comparisons made for differential expression should be included in the methods section. The FDR criteria in particular can be moved into the methods section completely for clarity in the results section of the texts

Author’s Response: We have now added more information in Materials and Methods (line 170–176): “DEGs can be identified using a false discovery rate (FDR) of < 0.05; however, we applied the rigorous criteria (FDR < 10-10) to identify genes that are differentially expressed in tenocytes following the application of tensile strain. Following criteria was used to identify DEGs: CON-A) DEGs among TT, MA, and AT samples were identified using the FDR of < 10-10 and CON-B) DEGs between TT and MA groups (designated as the TM group) were identified using the FDR of < 10-10 while FDR for AT was set at > 0.05.” 

5. The writing quality needs to be improved significantly throughout the manuscript. The Introduction and Discussion sections are extremely short and superficially written. They should be revised to include more detail about previous works and implications of these works both in context not just of their own work, but other studies in the tendon field.

Author’s Response: Thank you for your insightful suggestions. We have now extensively revised the Introduction and Discussion.

<Introduction>

Line 65–81: “Tendons are composed of cells, including tenocytes, and extracellular matrix (ECM). The most abundant ECM protein in the tendon is type I collagen, while other ECM components include proteoglycans and glycoproteins. Small proteoglycans regulate collagen assembly while large proteoglycans resist compressive forces. Furthermore, glycoproteins help in maintaining lubrication and elasticity of tendons. The largest tendon in the human body is the Achilles tendon, which inserts into the calcaneus and is defined as the distal confluence of the gastrocnemius and soleus muscles anatomically [1]. In general, tendons are capable of adapting to mechanical loading. Rupture of the Achilles tendon is frequently observed during physical activity due to mechanical overloading. Thus, proper mechanical loading is essential for structural integrity and functions of the tendon, and affects the metabolism of the tendon tissue. Mechanical forces (i.e. tension, fluid shear stress, and compression) exert various biological effects on the tendon. It has been shown that cyclic tensile strain affects collagen synthesis [2]. Whereas short-term cyclic tensile strain inhibits collagen production, long-term cyclic tensile strain promotes collagen synthesis [3]. In vitro, tenocytes from tendon explants exhibit increased DNA synthesis and enhanced collagen production in response to cyclic tensile strain, and increased sulfated glycosaminoglycan (GAG) content in response to static loading [4].”

<Discussion>

Line 253–258: “Although RNA-seq analysis of Achilles and patellar tendons have been performed in mice [23], studies comparing the gene expression profiles of Achilles and masticatory muscle tendons are sparse. Herein, we observed that the number of genes that were differentially regulated between AT and TT or between AT and MA were much more than the number of DEGs between MA and TT (S1 Table), suggesting that the characteristics of the Achilles tendon is different from that of the masticatory muscle tendon.”

Line 267–291: “The effects of cyclic tensile strain on tendon cells are well investigated. Although various strain amplitudes have been tested, most experiments used the frequency of 1.0 Hz, and very few studies have used the cyclic strain at low amplitude (1–3%). Lavagnino et al. reported that 1% static strain at a frequency of 1.0 Hz for 24 h inhibits matrix metalloproteinase (MMP)-1 expression in rat tail tendon fascicles in vitro [25]. Furthermore, a cyclic strain of 3% amplitude superimposed on a 2% static strain for 24 h enhances the retention of newly synthesized collagen into the fascicle of the rat tail tendon [3]. Another study by Lavagnino et al. [26] demonstrated that biaxial cyclic tensile strain of 12% amplitude at 1.0 Hz for 24 h significantly decreases actin depolymerization and increases collagenase expression. Based on these studies, we used cyclic sinusoidal equi-biaxial tensile strain at 1.0 Hz for 48 h from low (0%) to high (10%) amplitude in tendons isolated from the Japanese macaque. Maeda et al. performed microarray analysis of tenocytes exposed to cyclic strain [27]. In rat tail tendon fascicles, cyclic strain of 3% amplitude superimposed on a 2% static condition significantly reduces the expression of some MMP and ADAMTS family proteins while increases the expression of types VI and VIII collagen. This suggests that a cyclic strain of 3% amplitude superimposed on a 2% static strain for 24 h has anabolic effects on tenocytes. A previous study showed that ruptured Achilles tendons are surrounded by TUBB3 positive cells [21]. TUBB3 is expressed in non-neuronal tumor cells and plays a key role in regulating cell cycle progression [28]. Moreover, SEPTIN9 promotes cell migration and cytoskeletal rearrangements [29]. Our results showed that both TUBB3 and SEPTIN9 mRNA levels were decreased in stretched AT cells while their levels were increased in stretched MA and TT cells (Fig 6). We speculate that cyclic tensile strain induces cell cycle progression and cytoskeletal rearrangements in masticatory muscle tendon cells.”

Line 301–304: “Our study revealed that the expression of ADAMTS12, which prevents heterotopic ossification in tendons is down-regulated in tenocytes under tensile strain [34], which is consistent with the results of a previous microarray-based study on rat tail tendons under cyclic strain [27]”

Line 313–314: “These results suggest that cyclic sinusoidal equi-biaxial tensile strain has anabolic effects on tenocytes”

We have proofread our manuscript to improve the English language.

Minor comments:

Page 7, Line 16: The word “ingredient” is confusing here and should be removed. Perhaps rephrase the sentence to clearly indicate to the reader that tendon-specific trends in gene expression were analyzed with respect to stretching?

Author’s Response: Thank you for your valuable comments. We changed the sentence as follows (line 179–181): “To explore the differences in the gene expression pattern of masticatory tendons and Achilles tendons, we performed RNA-seq analysis of tenocytes isolated from these tendons”

Page 12, line 16: “MMATH” should be changed to “MMTAH”

Author’s Response: Thank you for pointing out the mistake. We have now corrected this mistake.

Page 12, line 16: The use of the word “primate” is preferred here. In addition, the authors should clarify whether they are referring to one species of primate or other monkeys of the same species.

Author’s Response: We have now edited this section for clarity (line 286–289): “In our previous study, we observed heterotopic calcification in tendons of the masticatory muscles in primates with MMTAH [9]; however, this study was performed in a single species.”. 

Following changes are also incorporated in the revised manuscript.

Gene expression profiling of the masticatory muscle tendons and Achilles tendons under tensile strain in the Japanese macaque Macaca fuscata

< Short title >

Strain-induced differential gene expression in Achilles tendons and tendons of masticatory muscles

<Abstract>

(line 55–59) “Moreover, the expression of tubulin beta 3 class III which promotes cell cycle progression, and septin 9 which promotes cytoskeletal rearrangements were decreased in stretched Achilles tendon cells while their expression was increased in stretched masseter aponeurosis and temporal tendon cells.”

<Materials and Methods>

(line 130) “~the masseter aponeurosis (MA), temporal tendon (TT), and Achilles tendon (AT) were dissected.”

<Results>

(line 198–201) “Following the application of tensile strain (48 h vs 0 h), we identified 2195 DEGs in the TT group, 1914 DEGs in the MA group, and 3697 DEGs in the AT group. We also identified 1076 DEGs on the stipulation of CON-B (S3 Table), and 147 DEGs (39 up-regulated, 108 down-regulated) in the TM group (Fig 3; S4 Table). ”

(line 236–243) “Next, we examined the one-to-one human orthologs of these 147 DEGs using the Ensemble database and identified 125 such orthologs (S5 Table). Among these orthologs, eight genes were related to tendons (Table 3), including sphingosine kinase 1 (SPHK1), alpha-2-macroglobulin (A2M), glycoprotein nmb (GPNMB), slit guidance ligand 3 (SLIT3), early growth response protein 1 (EGR1), ADAM metallopeptidase with thrombospondin type 1 motif 12 (ADAMTS12), CDK1, and follistatin-like 1 (FSTL1). The expression of GPNMB and SPHK1 was increased, while the expression of the other six genes was decreased in tenocytes following the application of tensile strain (Fig 6).”

“Fig 2. Experimental design and scheme for comparing different samples.”

“Fig 3. DEGs in the TM group. Venn diagram showing up- or down- regulated genes in the TM group. The number of up-regulated genes are less than that of the down-regulated genes.”

“Fig 4. Gene expression profile of each tendon sample. Heatmap with hierarchical clustering of the 452 DEGs in TT, MA, and AT samples (red, up-regulated; blue, down-regulated). Histogram shows the numerical value of each color.”

“Fig 5. Heatmap and boxplot showing the expression of 13 genes.

A) Heatmap with hierarchical clustering of 13 DEGs (red, up-regulated; blue, down-regulated). Histogram shows the numerical values of each color. B) Boxplot showing the relative expression of 13 selected DEGs.”

“Fig 6. Changes in the expression of the eight tendon-related genes after 48 h of exposure to tensile strain in AT, MA, and TT. The expression of GPNMB and SPHK1 is up-regulated whereas that of A2M, ADAMTS12, CDK1, EGR1, FSTL1, and SLIT3 is down-regulated in TT and MA following tensile strain.”

<Author Contributions>

“Funding Acquisition: Tsuyoshi Sato, Tetsuya Yoda and Masahiko Takada”

<Supporting Information>

“S1 Table. Differentially expressed genes in un-stretched TT, MA, and AT cells on the stipulation of CON-A.”

“S2 Table. Differentially expressed genes of the TM group on the stipulation of CON-B.”

“S3 Table. Differentially expressed genes with the condition of FDR < 10-10 in TT, MA, and AT.”

“S4 Table. Differentially expressed genes with the condition of FDR < 10-10 for TT and MA and FDR > 0.05 for AT.”

“S5 Table. List of human one-to-one orthologs differentially expressed only in TT and MA. Thirty-nine genes were up-regulated and one hundred eight genes were down-regulated”

---

## [Decision Letter · Decision Letter 1]

14 Dec 2022

PONE-D-22-06866R1Gene expression profiling of the masticatory muscle tendons and Achilles tendons under tensile strain in the Japanese macaque Macaca fuscataPLOS ONE

Dear Dr. Sato,

Thank you for submitting your manuscript to PLOS ONE. After careful consideration, we feel that it has merit but does not fully meet PLOS ONE’s publication criteria as it currently stands. Therefore, we invite you to submit a revised version of the manuscript that addresses the points raised during the review process. I have only one final minor request, before accepting the manuscript:As suggested by the reviewer could you please develop, in the discussion, the following point : "What might the differences mean in terms of clinical relevance or function?" Please submit your revised manuscript by Jan 28 2023 11:59PM. If you will need more time than this to complete your revisions, please reply to this message or contact the journal office at plosone@plos.org. Please include the following items when submitting your revised manuscript:A rebuttal letter that responds to each point raised by the academic editor and reviewer(s). You should upload this letter as a separate file labeled 'Response to Reviewers'.A marked-up copy of your manuscript that highlights changes made to the original version. You should upload this as a separate file labeled 'Revised Manuscript with Track Changes'.An unmarked version of your revised paper without tracked changes. You should upload this as a separate file labeled 'Manuscript'.If applicable, we recommend that you deposit your laboratory protocols in protocols.io to enhance the reproducibility of your results. Protocols.io assigns your protocol its own identifier (DOI) so that it can be cited independently in the future. For instructions see: https://journals.plos.org/plosone/s/submission-guidelines#loc-laboratory-protocols. Additionally, PLOS ONE offers an option for publishing peer-reviewed Lab Protocol articles, which describe protocols hosted on protocols.io. Read more information on sharing protocols at https://plos.org/protocols?utm_medium=editorial-email&utm_source=authorletters&utm_campaign=protocols.

We look forward to receiving your revised manuscript.

Kind regards,

Claude Prigent

Academic Editor

PLOS ONE

Journal Requirements:

Reviewers' comments:

Reviewer's Responses to Questions

**Comments to the Author**

1. If the authors have adequately addressed your comments raised in a previous round of review and you feel that this manuscript is now acceptable for publication, you may indicate that here to bypass the “Comments to the Author” section, enter your conflict of interest statement in the “Confidential to Editor” section, and submit your "Accept" recommendation.

Reviewer #2: (No Response)

2. Is the manuscript technically sound, and do the data support the conclusions?

Reviewer #2: Yes

3. Has the statistical analysis been performed appropriately and rigorously? 

Reviewer #2: I Don't Know

4. Have the authors made all data underlying the findings in their manuscript fully available?

Reviewer #2: Yes

5. Is the manuscript presented in an intelligible fashion and written in standard English?

Reviewer #2: Yes

6. Review Comments to the Author

Reviewer #2: The authors are to be commended on an interesting paper that demonstrates differences between functionally distinct load-bearing tendons. The authors have made significant improvements in this review however I found the discussion somewhat superficial and the meaning and relevance of the findings (aside from the fact that they are observationally different) in a wider context is not explored. What might the differences mean in terms of clinical relevance or function?

7. PLOS authors have the option to publish the peer review history of their article (what does this mean?). If published, this will include your full peer review and any attached files.

Reviewer #2: No

---

## [Author Response · Author response to Decision Letter 1]

31 Dec 2022

We thank the Editor and the Reviewers for their insightful and helpful comments. We have tried to address all issues raised by the reviewers. Also, we have thoroughly revised the manuscript. We highlighted the corrected or additional words/sentences with red font and the proofread words/sentences with the underline.

Review Comments to the Author

Reviewer #2: The authors are to be commended on an interesting paper that demonstrates differences between functionally distinct load-bearing tendons. The authors have made significant improvements in this review however I found the discussion somewhat superficial and the meaning and relevance of the findings (aside from the fact that they are observationally different) in a wider context is not explored. What might the differences mean in terms of clinical relevance or function?

Author’s Response: Thank you very much for the valuable comments. We have added the following sentences in the Discussion section (line 315-332): “Normally, the force applied to the masticatory tendon during chewing is approximately 320 N, and the force applied to the Achilles tendon during walking is approximately 1,000 N [38][39]. It is assumed that the Achilles tendon has high load-bearing properties. On the other hand, it has been reported that patients with MMTAH show hyperplasia of the masticatory muscle tendon without hyperplasia of the Achilles tendon, but the masticatory muscle tendon is subjected to a bite force of approximately 1,000 N [40]. Given these facts, it is consistent to assume that different genes are expressed in masticatory muscle tendons, which are considered vulnerable to high loads, than in Achilles tendons, which can withstand high loads. Although high loads were applied to each tendon cell in this study, different genes were expressed in the masticatory muscle tendon and Achilles tendon, respectively. In masticatory muscle tendon cells, which are vulnerable to high loads, the expression of genes that promote cell cycle progression and cytoskeletal reorganization (TUBB3 and SEPTIN9) and genes important for tendon regenerative capacity (SPHK1 and GPNMB) were elevated. We speculate that the faster turnover in highly loaded masticatory muscle tendon cells may increase the speed of cell regeneration, resulting in hyperplasia of tendon tissue. In other words, differences in gene expression between masticatory muscle tendons and Achilles tendons may be associated with differences in vulnerability to loading.”

We have added the following references.

[38] Lundgren D, Laurell L. Occlusal force pattern during chewing and biting in dentitions restored with fixed bridges of cross-arch extension. I. Bilateral end abutments. J Oral Rehabil. 1986 Jan;13(1):57-71.

[39] Fröberg A, Komi P, Ishikawa M, Movin T, Arndt A. Force in the achilles tendon during walking with ankle foot orthosis. Am J Sports Med. 2009 Jun;37(6):1200-7.

[40] Fujii T, Watanabe M, Motohashi T, Kubo H, Ohnishi Y, Shirao K, Nakajima M. Efficacy of surgical treatment for masticatory muscle tendon­aponeurosis hyperplasia. J Osaka Dent Univ 2020; 54 (2) : 213−218

---

## [Editor Report · Decision Letter 2]

5 Jan 2023

Gene expression profiling of the masticatory muscle tendons and Achilles tendons under tensile strain in the Japanese macaque Macaca fuscata

PONE-D-22-06866R2

Dear Dr. Sato,

We’re pleased to inform you that your manuscript has been judged scientifically suitable for publication and will be formally accepted for publication once it meets all outstanding technical requirements.

Kind regards,

Claude Prigent

Academic Editor

PLOS ONE
---

## [Editor Report · Acceptance letter]

9 Jan 2023

PONE-D-22-06866R2 

Gene expression profiling of the masticatory muscle tendons and Achilles tendons under tensile strain in the Japanese macaque *Macaca fuscata*

Dear Dr. Sato:

I'm pleased to inform you that your manuscript has been deemed suitable for publication in PLOS ONE. Congratulations! Your manuscript is now with our production department. 

Kind regards, 

on behalf of

Dr. Claude Prigent 

Academic Editor

PLOS ONE